# Object or Background: An Interpretable Deep Learning Model for COVID-19 Detection from CT-Scan Images

**DOI:** 10.3390/diagnostics11091732

**Published:** 2021-09-21

**Authors:** Gurmail Singh, Kin-Choong Yow

**Affiliations:** Faculty of Engineering and Applied Science, University of Regina, Regina, SK S4S 0A2, Canada; Gurmail.Singh@uregina.ca

**Keywords:** COVID-19, pneumonia, CT-scan, prototypical part

## Abstract

The new strains of the pandemic COVID-19 are still looming. It is important to develop multiple approaches for timely and accurate detection of COVID-19 and its variants. Deep learning techniques are well proved for their efficiency in providing solutions to many social and economic problems. However, the transparency of the reasoning process of a deep learning model related to a high stake decision is a necessity. In this work, we propose an interpretable deep learning model Ps-ProtoPNet to detect COVID-19 from the medical images. Ps-ProtoPNet classifies the images by recognizing the objects rather than their background in the images. We demonstrate our model on the dataset of the chest CT-scan images. The highest accuracy that our model achieves is 99.29%.

## 1. Introduction

The pandemic COVID-19 is looming as a worst menace on the world populations while its several new strains are being identified. Some vaccines for COVID-19 have been developed, but the list of the variants of COVID-19 is also getting bigger. There are seven lineages of the variants of the virus, such as: B.1.1.7, B.1.351, P.1, B.1.427, B.1.429, B.1.525, B.1.617.1 and B.1.617.2 [1]. The detection of the virus is usually done with molecular tests, that is, the tests that look for the virus by detecting the presence of the virus’s RNA. The molecular tests include RT-PCR, CRISPR, isothermal nucleic acid amplification, digital polymerase chain reaction, microarray analysis, and next-generation sequencing [2]. The presence of the virus can also be detected from the medical images, such as: chest *X*-ray and CT images. Although, RT-PCR is still a gold standard for COVID-19 testing, but deep learning techniques to identify the virus from medical images can also be helpful in certain circumstances, such as: unavailability of RT-PCR kits. A deep learning model can also be used for the pre-screening before RT-PCR testing. Many models have been proposed to detect COVID-19 from the medical images, see [3,4,5,6,7,8,9,10,11,12,13,14,15]. However, these models lack the interpretability/transparency of the reasoning process of their predictions. So, we propose an interpretable deep learning model: *pseudo prototypical part network* (Ps-ProtoPNet), and experiment it over the dataset of CT-scan images, see Section 2.4. Ps-ProtoPNet is closely related to ProtoPNet [16], Gen-ProtoPNet [17] and NP-Proto-PNet [18], but strikingly different from these models.

A prototype represents a patch of an image. To classify a test image, ProtoPNet compares the different parts of the test image with the learned prototypes of images from all classes. Then the decision is made based on the weighted combination of similarity scores [16]. To calculate the similarity scores between learned prototypes (with square spatial dimensions 1×1) and parts of the test image, ProtoPNet and NP-ProtoPNet use L2 distance function, whereas Gen-ProtoPNet uses a generalized version of L2.

In this work, we present a theorem that calculates the impact of the change in the hyperparameters of the dense layer on the logits, see Theorem 1. Ps-ProtoPNet chooses negative connections between the similarity score and logits of incorrect classes as suggested by the theorem. Also, our model uses prototypes that can have any type of spatial dimensions, that is, square and rectangular.

A model should classify an image of an *object* by identifying the object in the image instead of the *background* of the object in the image. The model that uses prototypes of smaller spatial dimensions (1×1) can classify an image just on the basis of the background and give higher accuracy with wrong reasoning process. For example, the most part of the images of birds of a sea specie is not similar to any patch of the images of birds of a jungle specie. So, the images from these two classes can be classified on the basis of backrounds. Another scenario, images of birds of different sea bird species can share same background water on the most part. Therefore, a model with prototypes of small spatial dimensions (1×1) can classify wrongly the images just on the basis of the background of the birds. On the other hand, the use of prototypes with the dimensions equal to the dimensions of an image can also reduce the accuracy because there can be only few images that are similar to the whole image, but their parts can be similar. So, we need to use optimum spatial dimensions for the prototypes. To identify an image that has not been encountered before, humans may compare patches of the image with the patches of images of the known objects. Our model’s reasoning is inspired from the above reasoning, where comparison of image parts with learned prototypes is integral to the reasoning process of the model. That is, a new image is compared with learned prototypes from all classes, and it is classified to the class whose prototypes are more similar to parts of the image. We have three classes of images: Covid, Normal and Pneumonia. Therefore, a COVID-19 CT image is distinguished from the pneumonia CT images based on the greater similarity of parts of the image with the prototypes.

## 2. Materials and Methods

### 2.1. Related Work

Numerous perspectives have been emerged to explain convolution neural networks, including posthoc interpretability analysis. A neural network with *posthoc* analysis is interpreted following classifications made by a model. Activation maximization [19,20,21,22,23,24,25], deconvolution [26], and saliency visualization [23,27,28,29] are some forms of posthoc analysis approach. Nevertheless, these techniques do not throw light on the reasoning process with transparency. Another approach to make the reasoning process of the neural networks clear is attention-based interpretability that includes class activation maps (CAM) and part-based models. In this approach, a model aims to point out the parts of a test image that are its centers of attention [30,31,32,33,34,35,36,37,38,39,40,41]. These models do not point out the prototypes that are similar to parts of the test image.

Oscar et al. [42] developed a model that uses prototypes of the size of a whole image to find the similarity scores. A substantial improvement over the above work was made by Chen et al. with the development of their model ProtoPNet [16]. The models Gen-ProtoPNet [17] and NP-ProtoPNet [18] are close variations of ProtoPNet.

### 2.2. Data

Many datasets of medical images are publicly available [43,44,45]. However, we used the dataset of chest CT-scan images of normal people, COVID-19 patients and pneumonia patients [44]. This dataset has 143,778 training images and 25,658 test images. The training dataset consists of 35,996, 25,496 and 82,286 CT-scan images of normal people, pneumonia patients and COVID-19 patients, respectively. The test dataset consists of 12,245, 7395 and 6018 CT-scan images of normal people, pneumonia patients and COVID-19 patients. We resized the images to the dimensions 224×224 as required by the base models. We put these images into three classes Covid (first class), Normal (second class) and Pneumonia (third class).

### 2.3. Working Principal and Novelty of Ps-ProtoPNet

ProtoPNet classify an image on the basis of a weighted combination of the similarity scores [16]. For each class, a fixed number of prototypes are selected. We select 10 prototypes for each class. The model calculates the Euclidean distance of each prototype from each latent patch of the test image that has spatial dimensions equal to 1×1. Then these distances are inverted and a maximum of the inverted distances is called the similarity score of the prototype. Thus, for a given image, only one similarity score for each prototype is obtained. In the dense layer, these similarity scores are multiplied with the weights to calculates the logits. During the training process, ProtoPNet does the convex optimization of the last layer to make the certain weights zero [16].

Theorem 1 finds the impact of the change in the weights on the logits. Therefore, along with the use of prototypes with spatial dimensions bigger than 1×1, Ps-ProtoPNet uses the negative weights for similarity scores that connect to incorrect classes. Thus, for a given CT-scan image as in Figure 1, Ps-ProtoPNet identifies the parts of the image where it thinks that this part of the image looks like that prototypical part, and this part of the image does not look like that prototypical part. In addition to the positive reasoning process, Ps-ProtoPNet does not do the convex optimiza-tion of the last layer to keep the impact of the negative reasoning process on the image classification, whereas ProtoPNet model emphasizes on the positive reasoning process. The non-optimization of the last layer enabled us to write Theorem 1, because it ensures that the weights of last layer do not change during the training process. Also, it reduces the training time considerably.

### 2.4. Ps-ProtoPNet Architecture

In this section, we introduce and explain the architecture and the training procedure of our model Ps-ProtoPNet in the context of CT-scan images.

We construct our network over the state-of-the-art mod-els: VGG-16, VGG-19 [46], ResNet-34, ResNet-152 [47], DenseNet-121, or DenseNet-161 [48]. In this paper, these models are called baseline or base models. The base models were pretrained on ImageNet [49]. In the Figure 2, we see that the model comprises of the convolution layers of any of the above base model that are followed by an additional 1×1 layer (we denote these convolution layers together by *ℓ*) and then these convolution layers are followed by a generalized [50,51] convolution layer $p_{p}$ of prototypical parts and a dense layer *w* with weight matrix $m_{w}$. The dense layer does not have any bias. We denote the parameters of *ℓ* by $\ell_{conv}$. The activation function Sigmoid is used for the additional convolution layer.

We provide an explanation of our model with the base model VGG-16. For an input image *x*, let ℓ(x) be the output of the convolutional layers *ℓ*. Therefore, the shape of ℓ(x) is 512×7×7. Let $P^{k}={\left\{p_{l}^{k}\right\}}_{l=1}^{m^{\prime }}$ be the set of prototypes of a class *k* and $P={\left\{P^{k}\right\}}_{k=1}^{n}$ is set of prototypes of all classes, where $m^{\prime }$ is the number of prototypes for each class and *n* is the total number of classes. In our case, $m^{\prime }=10$ and n=3, and the hyperparameter $m^{\prime }=10$ is chosen randomly. For example, $p_{1}^{1},p_{2}^{1},\ldots ,p_{10}^{1}$ prototypes belong to the first class (Covid class). The shape of each prototype is 512×h×w, where 1×1<h×w<7×7, that is, *h* and *w* are neither simultaneously equal to 1 nor 7. Hence, every prototype can be considered as a representation of some prototypical part of some CT-scan image.

As explained in Section 2.3, Ps-ProtoPNet calculates the similarity scores between an input image and the prototypical parts $p_{1}^{1}-p_{10}^{1}$, $p_{1}^{2}-p_{10}^{2}$ and $p_{1}^{3}-p_{10}^{3}$, see Figure 2. Note that, similarity score of the prototype $p_{1}^{1}$ (0.03955200) is greater than the similarity scores of $p_{1}^{2}$ (0.00021837) and $p_{10}^{3}$ (0.00023386). The complete list is given in the similarity score matrix *S*, see Section 2.6. The source image of the prototypes $p_{1}^{1}$, $p_{1}^{2}$ and $p_{10}^{3}$ are also given in the third column of the Figure 2. The model keeps track of spatial relation of the convolutional output and the prototypical parts, and upsamples the parts to the size of input image to point out the patch on the source images that corresponds to the prototypes. The rectangles in the source images are the parts of the source images from where the prototypical parts are taken. In layer *w*, the matrix *S* is multiplied with $m_{w}$ to get the logits. The logits for the first, second and third class are 0.5744, −0.5790 and −0.5787, respectively.

### 2.5. The Training of Ps-ProtoPNet

We use the generalized version *d* of the distance function L2 (Euclidean distance). We consider the baseline VGG-16 to present *d* in this section. For a given image *x*, let z=ℓ(x). Therefore, the shape of ℓ(x) is 512×7×7, where 512 is the depth of ℓ(x) and 7×7 are the spatial dimensions of ℓ(x). Let *p* be a prototype of the shape 512×h×w, where 1≤h,w≤7, but *h* and *w* are neither simultaneously equal to 1 nor 7. Since *p* can be any prototype of any class, *p* does not have any subscript and superscript. The output *z* of the convolutional layers *ℓ* has (8−h)(8−w) patches of dimensions h×w. Hence, square of the distance $d(Z_{ij},p)$ between the prototype *p* and (i,j) patch $Z_{ij}$ (say) of *z* is:(1)$$d^{2}\left(Z_{ij},p\right)=\sum_{l=1}^{h}\sum_{m=1}^{w}\sum_{k=1}^{512}\left|\right|z_{(i+l-1)(j+m-1)k}-p_{lmk}{\left|\right|}_{2}^{2}.$$

For prototypes of spatial dimension 1×1, that is, h=w=1, we have $d^{2}\left(Z_{ij},p\right)=\sum_{k=1}^{512}\left|\right|z_{ijk}-p_{11k}{\left|\right|}_{2}^{2}$, which is the square of the Euclidean distance between the prototype *p* and a patch of *z*, where $p_{11k}≃p_{k}$. Therefore, the distance function *d* is a generalization of L2. The prototypical unit $p_{p}$ calculates the following.
$$p_{p}\left(z\right)=\max_{1\leq i\leq 8-h,\;1\leq j\leq 8-w}\log\left(\frac{d^{2}\left(Z_{ij},p\right)+1}{d^{2}\left(Z_{ij},p\right)+\epsilon }\right).$$

In other words,
(2)$$p_{p}\left(z\right)=\max_{Z\in patches(z)}\log\left(\frac{d^{2}\left(Z,p\right)+1}{d^{2}\left(Z,p\right)+\epsilon }\right).$$

The Equation (Equation 2) tells us that a prototype *p* is more similar to input image *x* if the inverse of the distance between a latent patch of *x* and *p* is smaller. The two training steps of our model are as follows.

#### 2.5.1. Optimization of All Layers before the Dense Layer

Suppose $X=\{x_{1}\ldots x_{n}\}$ and $Y=\{y_{1}\ldots y_{n}\}$ are sets of images and corresponding labels, respectively. Let $D=\{\left(x_{i},y_{i}\right):x_{i}\in X,y_{i}\in Y\}$. Our objective function is:(3)$$\min_{P,\ell_{conv}}\frac{1}{n}\sum_{i=1}^{n}CrosEnt\left(h\circ p_{p}\circ \ell \left(x_{i}\right),y_{i}\right)+\lambda_{1}ClstCst+\lambda_{2}SepCst,$$
where ClstCst and SepCst are:(4)$$ClstCst=\frac{1}{n}\sum_{i=1}^{n}\min_{j:p_{j}\in P_{y_{i}}}\min_{Z\in patches(\ell \left(x_{i}\right))}d^{2}\left(Z,p_{j}\right);$$
(5)$$SepCst=-\frac{1}{n}\sum_{i=1}^{n}\min_{j:p_{j}\notin P_{y_{i}}}\min_{Z\in patches(\ell \left(x_{i}\right))}d^{2}\left(Z,p_{j}\right).$$

The Equation (Equation 4) tells us that the decrease in the cluster cost (ClstCst) leads to clustering of prototypes surrounding their respective classes. However, the Equation (Equation 5) suggests that the decrease in separation cost (SepCst) keeps prototypes away from their incorrect classes [16]. The drop in cross entropy leads to improved classifications, see the objective function (Equation 3). The hyperparameters $\lambda_{1}$ and $\lambda_{2}$ are selected from the set {0.4,0.5,0.7,0.8,0.9} using cross validation. Since $m_{w}$ is the weight matrix for the last layer, $m_{w}^{\left(i,j\right)}$ is the weight assigned to the connection between similarity score of *j*th prototype and logit of *i*th class. Theorem 1 finds the impact of the selection of the weights $m_{w}^{\left(i,j\right)}$ on the logits. Therefore, for a class *k*, we put $m_{w}^{\left(i,j\right)}=1$ for all *j* with $p_{j}^{i}\in P^{i}$, and for all $p_{j}^{k}\notin P^{i}$ with k≠i, $m_{w}^{\left(k,j\right)}$ is chosen from the set {−1,−0.9,−0.7,−0.5,−0.2,−0.1}. Since the distance function is nonnegative, the optimization of all layers except the last layer with the optimizer SGD helps Ps-ProtoPNet to learn important latent space.

#### 2.5.2. Push of Prototypical Parts

At this step, Ps-ProtoPNet pushes/projects the prototypes onto the patches of the output ℓ(x) of an image *x* that have smallest distances from the prototypes. That is, Ps-ProtoPNet performs the following update:$$p_{j}^{k}⟵arg\min_{\left\{Z:Z\in patches\left(\ell \left(x_{i}\right)\right)\;\forall i\;s.t.\;y_{i}=k\right\}}d\left(Z,p_{j}^{k}\right).$$

Therefore, prototype layer gets updated prototypical parts that are more closer to their respective classes [16]. The patch of *x* that is the most similar to *p* is used for visualization of *p*. The activation value of the prototype must be at least 94th percentile of all the activation values of $p_{p}$ [16].

### 2.6. Explanation of Ps-ProtoPNet with an Example

The test image in the first column of Figure 3 belongs to the first class (Covid). In the second column, the test image has some patches enclosed in green rectangles. These patches give the highest similarity score to the corresponding prototypes in the third column. The prototypes in the third column are taken from the corresponding source images in the fourth column. The rectangles on the source image pin-point the patches from where the corresponding prototypes are taken. The fifth column has similarity scores of the prototypes and sixth column has the weights. The entries of the seventh column are obtained by multiplying the similarity scores with the corresponding weights. The logit (0.5744) of the first class is the sum of entries of the seventh column. The logit for the first class can also be obtained from the multiplication of the first row of weight matrix $m_{w}$ with the similarity score matrix *S*. Similarly, the logit for the second class (−0.5790) and third class (−0.5787) can be obtained by multiplying second and third row of the weight matrix with the similarity score matrix *S*.

The transpose of the weight matrix $m_{w}$ and similarity scores matrix *S* that we obtain from our experiments are as follows:$$m_{w}^{T}=\left[\begin{array}{ccc} 1 & -1 & -1\\ 1 & -1 & -1\\ 1 & -1 & -1\\ 1 & -1 & -1\\ 1 & -1 & -1\\ 1 & -1 & -1\\ 1 & -1 & -1\\ 1 & -1 & -1\\ 1 & -1 & -1\\ 1 & -1 & -1\\ -1 & 1 & -1\\ -1 & 1 & -1\\ -1 & 1 & -1\\ -1 & 1 & -1\\ -1 & 1 & -1\\ -1 & 1 & -1\\ -1 & 1 & -1\\ -1 & 1 & -1\\ -1 & 1 & -1\\ -1 & 1 & -1\\ -1 & -1 & 1\\ -1 & -1 & 1\\ -1 & -1 & 1\\ -1 & -1 & 1\\ -1 & -1 & 1\\ -1 & -1 & 1\\ -1 & -1 & 1\\ -1 & -1 & 1\\ -1 & -1 & 1\\ -1 & -1 & 1 \end{array}\right]\;\mathrm{and}\;S=\left[\begin{array}{c} 0.03955200\\ 0.03955200\\ 0.03955200\\ 0.03955200\\ 0.03955200\\ 0.03955200\\ 0.03955200\\ 0.03955200\\ 0.22292000\\ 0.03955200\\ 0.00021837\\ 0.00021837\\ 0.00021837\\ 0.00021837\\ 0.00021837\\ 0.00021837\\ 0.00021837\\ 0.00021837\\ 0.00021837\\ 0.00021813\\ 0.00023386\\ 0.00023386\\ 0.00023374\\ 0.00023386\\ 0.00023386\\ 0.00023386\\ 0.00023386\\ 0.00023386\\ 0.00023374\\ 0.00023386 \end{array}\right].$$

## 3. Results

In this section we present the metrics given by our model and compare the performance of our model with the performance of the other models.

### 3.1. The Metrics and Confusion Matrices

For a given class, true positive (*TP*) and true negative (*TN*) are the number of items correctly predicted as belonging to the class and not belonging to the class, respectively, see [52]. False positives (*FP*) and false negatives (*FN*) are the number of items incorrectly predicted as belonging to the class and not belonging to the class, respectively, see [53]. The metrics accuracy, precision, recall and F1-score are [54,55,56]:(6)$$\;\begin{array}{ccc} & \; & Accuracy=\frac{TP+TN}{Total\;Cases},\;Precision=\frac{TP}{TP+FP}. \end{array}$$
(7)$$\;\begin{array}{ccc} & \; & Recall=\frac{TP}{TP+FN},\;F1-score=\frac{2}{{Precision}^{-1}+{Recall}^{-1}}. \end{array}$$

In Figure 4, Figure 5, Figure 6, Figure 7, Figure 8 and Figure 9, the confusion matrices of Ps-ProtoPNet with the base models are given. For example, in Figure 4, the confusion matrix *N* (say) of Ps-ProtoPNet with base model VGG-16 is provided. Thus, the numbers N[0][0], N[1][1]+N[2][2], N[0][1]+N[0][2] and N[1][0]+N[2][0] denote the true positives *TP*, true negatives *TN*, false positives *FP* and false negatives *FN* of the Covid class. Therefore, by Equations (Equation 6) and (Equation 7), the accuracy for Ps-ProtoPNet is 98.83%, and the precision, recall and F1-score are equal to 0.96, 0.98 and 0.97, respectively.

### 3.2. The Performance Comparison of the Models

The models Ps-ProtoPNet, Gen-ProtoPNet, NP-ProtoPNet and ProtoPNet are constructed over the convolution layers of base models. We trained and tested these models over the dataset of CT-scan images [44]. Although, the accuracies of these models stabilize before 30 epochs (see Section 3.3), but we trained and tested the models for 100 epochs.

The comparison of the performance in the metrics is given in the Table 1. We observe from the third column of Table 1 that when we construct our model over the convolutional layers of VGG-16, and use the prototypes of spatial dimensions 3×4 then the accuracy, precision, recall and F1-score given by Ps-ProtoPNet are 98.83, 0.96, 0.98 and 0.97, respectively. The accuracy, precision, recall and F1-score given by the models Gen-ProtoPNet, NP-ProtoPNet and ProtoPNet with baseline VGG-16 are 95.85, 0.93, 0.95 and 0.94; 98.23, 0.93, 0.95 and 0.94; and 90.84, 0.89, 0.91 and 0.90, respectively. The accuracy, precision, recall and F1-score given by VGG-16 itself (Base only) are 99.03, 0.98, 0.99 and 0.98, respectively. Also, we observe from the Table 1 that the performance of Ps-ProtoPNet is the highest after base models.

### 3.3. The Graphical Comparison of the Accuracies

In the Figure 10, Figure 11, Figure 12, Figure 13, Figure 14 and Figure 15, the accuracies given by Ps-ProtoPNet are graphically compared with the accuracies given by the other models. As mentioned in Section 3.2, the accuracies of these models stabilize before 30 epochs, but we trained and tested the models for 100 epochs over the dataset of CT-scan images [44]. In Figure 10, the comparison of the accuracies given by the models with baseline VGG-16 is provided. The curves of colors green, purple, yellow, brown and blue sketch the accuracies of Ps-ProtoPNet, Gen-ProtoPNet, NP-ProtoPNet, ProtoPNet and VGG-16, respectively. Although, it is hard to see the difference between the accuracies in the Figure 10, Figure 11, Figure 12, Figure 13, Figure 14 and Figure 15, but the figures clearly show the difference between the accuracies before they stabilize.

### 3.4. The Test of Hypothesis for the Accuracies

Since accuracy is the proportion of correctly classified images among all the test images, we can apply the test of hypothesis concerning system of two proportions. Let *n* be the size of test dataset, and the number of images correctly classified by model 1 and 2 are $x_{1}$ and $x_{2}$, respectively. Let ${\tilde{p}}_{1}=x_{1}/n$ and ${\tilde{p}}_{2}=x_{2}/n$. The statistic for test concerning difference between two proportions is given by [57]:(8)$$Z=\frac{{\tilde{p}}_{1}-{\tilde{p}}_{2}}{\sqrt{2\tilde{p}\left(1-\tilde{p}\right)/n}},\;where\;\tilde{p}=\left(x_{1}+x_{2}\right)/2n.$$

Let $p_{1}$ and $p_{2}$ be the accuracies given by model 1 and 2. Therefore, our hypothesis is as follows:

$H_{0}:\left(p_{1}-p_{2}\right)=0$ (null hypothesis)

$H_{a}:\left(p_{1}-p_{2}\right)\neq 0$ (alternative hypothesis)

We test the hypothesis for the level of confidence (α) = 0.05. Since the hypothesis is two-tailed, the *p*-value must be less than 0.025 to reject the null hypothesis. In the above hypotheses, $p_{1}$ is the accuracy given by Ps-ProtoPNet and $p_{2}$ represents the accuracies given by Gen-ProtoPNet, NP-ProtoPNet, ProtoPNet and base models. We obtain the values of test statistic *Z* from the above formula given by the Equation (Equation 8). Then the corresponding *p*-values are obtained from the standard normal table (Z-table). The complete list of *p*-values is given in the Table 2. For example, when VGG-16 is used as a base model, the *p*-values obtained from the accuracy given by Gen-ProtoPNet in pairs with accuracies given by Gen-ProtoPNet, NP-ProtoPNet and ProtoPNet are 0.0002,0.0002,0.0002 and 0.0367, respectively. Since α=0.05, we reject the null hypothesis for all the *p*-values listed in the Table 2 except the five *p*-values written in bold. The *p*-values in bold in the last column means the accuracies given by Ps-ProtoPNet are not statistically different from accuracies given by the three base models. However, we can say with 95% confidence that the accuracies given by Ps-ProtoPNet are better than the corresponding accuracies given by Gen-ProtoPNet, NP-ProtoPNet and ProtoPNet except in the two cases.

### 3.5. The Impact of Change in the Hyperparameters of the Last Layer

In this section, we prove a theorem analogous to [16], Theorem 2.1. Our experiments show that $w_{m}^{\left(k,j\right)}$ can hardly be made equal to 0 for $p_{j}^{k}\notin P^{i}$ during the training, an assumption made in [16], Theorem 2.1. Therefore, we don’t assume this condition.

**Theorem** **1.**
*Let $h\circ p_{p}\circ \ell$ be a Ps-ProtoPNet. For a class k, let $b_{l}^{k}$ and $a_{l}^{k}$ be the values of l-th prototype for class k before the projection of $p_{l}^{k}$ and after the projection of $p_{l}^{k}$, respectively. Let x be an input image that is correctly classified by Ps-ProtoPNet before the projection, and k be the correct class label of x. Suppose that:*
*A1* 
*$z_{l}^{k}=arg\min_{\tilde{z}\in patches\left(\ell \left(x\right)\right)}d\left(\tilde{z},a_{l}^{k}\right)$;*
*A2* 
*there exists some δ with 0<δ<1 such that:*
*A2a* 
*for all incorrect classes $k^{\prime }\neq k$ and $l\in \{1,\ldots ,m_{k^{\prime }}\}$, we have $d\left(a_{l}^{k^{\prime }},b_{l}^{k^{\prime }}\right)\leq \theta d\left(z_{l}^{k^{\prime }},b_{l}^{k^{\prime }}\right)-\sqrt{\epsilon }$, where ϵ is given by $p_{p}\left(z\right)=\max_{Z\in patches(z)}\log\left(\frac{d^{2}\left(Z,p\right)+1}{d^{2}\left(Z,p\right)+\epsilon }\right)$ and $\theta =\min(\sqrt{1+\delta }-1,1-\frac{1}{\sqrt{2-\delta }})$;*
*A2b* 
*for all $l\in \{1,\ldots ,m_{k}\}$, we have $d\left(a_{l}^{k},b_{l}^{k}\right)\leq \left(\sqrt{1+\delta }-1\right)d\left(z_{l}^{k},b_{l}^{k}\right)\;and\;d\left(z_{l}^{k},b_{l}^{k}\right)\leq \sqrt{1-\delta }.$*



*Then after projection, the output logit for the correct class k can decrease at most by $\Delta =m^{\prime }\log\left(1+\delta \right)\left(2-\delta \right)\left(1+\frac{1}{r}\left(n-1\right)\right)$, where −1/r is the weight assigned to incorrect classes, and r is a positive real number.*


**Proof of Theorem 1.** For any class *k*, let $L_{k}\left(x,{\left\{p_{l}^{k}\right\}}_{l=1}^{m^{\prime }}\right)$ be the output logit for input image *x*, where ${\left\{p_{l}^{k}\right\}}_{l=1}^{m^{\prime }}$ denote the prototypes of class *k*. Since negative connections between similarities scores of incorrect classes and logits are equal to −1/r,

$\begin{array}{ccc} L_{k}\left(x,{\left\{p_{l}^{k}\right\}}_{l=1}^{m^{\prime }}\right) & = & \sum_{l=1}^{m^{\prime }}\log\left(\frac{d^{2}\left(z_{l}^{k},p_{l}^{k}\right)+1}{d^{2}\left(z_{l}^{k},p_{l}^{k}\right)+\epsilon }\right)-\frac{1}{r}\sum_{k^{\prime }\neq k}\sum_{l=1}^{m^{\prime }}\log\left(\frac{d^{2}\left(z_{l}^{k^{\prime }},p_{l}^{k^{\prime }}\right)+1}{d^{2}\left(z_{l}^{k^{\prime }},p_{l}^{k^{\prime }}\right)+\epsilon }\right). \end{array}$

Let $\Delta_{k}$ be the difference between the output logit of class *k* before and after the projection of prototypes ${\left\{p_{l}^{k}\right\}}_{l=1}^{m^{\prime }}$ to their nearest latent training patches. Suppose $L_{k}\left(x,{\left\{b_{l}^{k}\right\}}_{l=1}^{m^{\prime }}\right)$ and $L_{k}\left(x,{\left\{a_{l}^{k}\right\}}_{l=1}^{m^{\prime }}\right)$ denotes the logits before the projection and after the projection, respectively. Therefore, we have
$$\begin{array}{ccc} \;\Delta_{k} & \;= & \;L_{k}\left(x,{\left\{a_{l}^{k}\right\}}_{l=1}^{m^{\prime }}\right)-L_{k}\left(x,{\left\{b_{l}^{k}\right\}}_{l=1}^{m^{\prime }}\right)\\ & \;= & \;\sum_{l=1}^{m^{\prime }}\log\left(\frac{d^{2}\left(z_{l}^{k},a_{l}^{k}\right)+1}{d^{2}\left(z_{l}^{k},b_{l}^{k}\right)+1}\cdot \frac{d^{2}\left(z_{l}^{k},b_{l}^{k}\right)+\epsilon }{d^{2}\left(z_{l}^{k},a_{l}^{k}\right)+\epsilon }\right)-\frac{1}{r}\sum_{k^{\prime }\neq k}\sum_{l=1}^{m^{\prime }}\log\left(\frac{d^{2}\left(z_{l}^{k^{\prime }},a_{l}^{k^{\prime }}\right)+1}{d^{2}\left(z_{l}^{k^{\prime }},b_{l}^{k^{\prime }}\right)+1}\cdot \frac{d^{2}\left(z_{l}^{k^{\prime }},b_{l}^{k^{\prime }}\right)+\epsilon }{d^{2}\left(z_{l}^{k^{\prime }},a_{l}^{k^{\prime }}\right)+\epsilon }\right). \end{array}$$Suppose that,
(9)$$\begin{array}{c} \Psi_{l}^{k}=\frac{d^{2}\left(z_{l}^{k},a_{l}^{k}\right)+1}{d^{2}\left(z_{l}^{k},b_{l}^{k}\right)+1}\times \frac{d^{2}\left(z_{l}^{k},b_{l}^{k}\right)+\epsilon }{d^{2}\left(z_{l}^{k},a_{l}^{k}\right)+\epsilon },and\;\Psi_{l}^{k^{\prime }}=\frac{d^{2}\left(z_{l}^{k^{\prime }},a_{l}^{k^{\prime }}\right)+1}{d^{2}\left(z_{l}^{k^{\prime }},b_{l}^{k^{\prime }}\right)+1}\times \frac{d^{2}\left(z_{l}^{k^{\prime }},b_{l}^{k^{\prime }}\right)+\epsilon }{d^{2}\left(z_{l}^{k^{\prime }},a_{l}^{k^{\prime }}\right)+\epsilon }. \end{array}$$Therefore,
(10)$$\Delta_{k}=\sum_{l=1}^{m^{\prime }}\log\Psi_{l}^{k}-\sum_{k^{\prime }\neq k}\sum_{l=1}^{m^{\prime }}\log\Psi_{l}^{k^{\prime }}.$$From the inequality given in the assumption (A2b), we have
(11)$$\frac{d^{2}\left(z_{l}^{k},a_{l}^{k}\right)+1}{d^{2}\left(z_{l}^{k},b_{l}^{k}\right)+1}\geq \frac{1}{d^{2}\left(z_{l}^{k},b_{l}^{k}\right)+1}\geq \frac{1}{2-\delta }.$$By the triangle inequality, we have $d\left(z_{l}^{k},a_{l}^{k}\right)\leq d\left(z_{l}^{k},b_{l}^{k}\right)+d\left(a_{l}^{k},b_{l}^{k}\right).$ Consequently,
(12)$$\frac{d^{2}\left(z_{l}^{k},b_{l}^{k}\right)+\epsilon }{d^{2}\left(z_{l}^{k},a_{l}^{k}\right)+\epsilon }\geq \frac{d^{2}\left(z_{l}^{k},b_{l}^{k}\right)+\epsilon }{{\left(d\left(z_{l}^{k},b_{l}^{k}\right)+d\left(a_{l}^{k},b_{l}^{k}\right)\right)}^{2}+\epsilon }.$$Again, by (A2b), we have
$$d\left(a_{l}^{k},b_{l}^{k}\right)\leq \left(\sqrt{1+\delta }-1\right)d\left(z_{l}^{k},b_{l}^{k}\right),thatis,\;d\left(a_{l}^{k},b_{l}^{k}\right)+d\left(z_{l}^{k},b_{l}^{k}\right)\leq d\left(z_{l}^{k},b_{l}^{k}\right)\sqrt{1+\delta }.$$On squaring both sides of the above inequality and then adding ϵ to both sides of the inequality, we obtain
$$\begin{array}{ccc} {\left(d\left(a_{l}^{k},b_{l}^{k}\right)+d\left(z_{l}^{k},b_{l}^{k}\right)\right)}^{2}+\epsilon & \leq & \left(1+\delta \right)d^{2}\left(z_{l}^{k},b_{l}^{k}\right)+\epsilon \leq \left(1+\delta \right)\left(d^{2}\left(z_{l}^{k},b_{l}^{k}\right)+\epsilon \right). \end{array}$$On rearranging the above inequality, we have
(13)$$\frac{d^{2}\left(z_{l}^{k},b_{l}^{k}\right)+\epsilon }{{\left(d\left(a_{l}^{k},b_{l}^{k}\right)+d\left(z_{l}^{k},b_{l}^{k}\right)\right)}^{2}+\epsilon }\geq \left(1+\delta \right).$$Therefore, by inequalities (Equation 12) and (Equation 13), be obtain
(14)$$\frac{d^{2}\left(z_{l}^{k},b_{l}^{k}\right)+\epsilon }{d^{2}\left(z_{l}^{k},a_{l}^{k}\right)+\epsilon }\geq \frac{(d^{2}\left(z_{l}^{k},b_{l}^{k}\right)+\epsilon }{{\left(d\left(a_{l}^{k},b_{l}^{k}\right)+d\left(z_{l}^{k},b_{l}^{k}\right)\right)}^{2}+\epsilon }\geq \left(1+\delta \right).$$Hence, by Equations (Equation 11) and (Equation 14), we have
(15)$$\Psi_{l}^{k}=\frac{d^{2}\left(z_{l}^{k},a_{l}^{k}\right)+1}{d^{2}\left(z_{l}^{k},b_{l}^{k}\right)+1}\times \frac{d^{2}\left(z_{l}^{k},b_{l}^{k}\right)+\epsilon }{d^{2}\left(z_{l}^{k},a_{l}^{k}\right)+\epsilon }\geq \frac{1}{\left(1+\delta )(2-\delta \right)}.$$Now we derive an upper bound of $\Psi_{l}^{k^{\prime }}$, where $k^{\prime }\neq k$. Using the triangle inequality, we obtain
$$d^{2}\left(z_{l}^{k^{\prime }},a_{l}^{k^{\prime }}\right)\leq {\left(d\left(z_{l}^{k^{\prime }},b_{l}^{k^{\prime }}\right)+d\left(a_{l}^{k^{\prime }},b_{l}^{k^{\prime }}\right)\right)}^{2}+1.$$Therefore,
(16)$$\frac{d^{2}\left(z_{l}^{k^{\prime }},a_{l}^{k^{\prime }}\right)+1}{d^{2}\left(z_{l}^{k^{\prime }},b_{l}^{k^{\prime }}\right)+1}\leq \frac{{\left(d\left(z_{l}^{k^{\prime }},b_{l}^{k^{\prime }}\right)+d\left(a_{l}^{k^{\prime }},b_{l}^{k^{\prime }}\right)\right)}^{2}+1}{d^{2}\left(z_{l}^{k^{\prime }},b_{l}^{k^{\prime }}\right)+1}.$$By assumption (A2a), we have
(17)$$\begin{array}{ccc} d(a_{l}^{k^{\prime }},b_{l}^{k^{\prime }}) & \leq & \left(\sqrt{1+\delta }-1\right)d\left(z_{l}^{k^{\prime }},b_{l}^{k^{\prime }}\right)-\sqrt{\epsilon }\leq \left(\sqrt{1+\delta }-1\right)d\left(z_{l}^{k^{\prime }},b_{l}^{k^{\prime }}\right). \end{array}$$By the inequality (Equation 17), we have
(18)$$\begin{array}{ccc} {\left(d\left(z_{l}^{k^{\prime }},b_{l}^{k^{\prime }}\right)+d\left(a_{l}^{k^{\prime }},b_{l}^{k^{\prime }}\right)\right)}^{2}\; & \leq & {\left(d\left(z_{l}^{k^{\prime }},b_{l}^{k^{\prime }}\right)+\left(\sqrt{1+\delta }-1\right)d\left(z_{l}^{k^{\prime }},b_{l}^{k^{\prime }}\right)\right)}^{2}\\ & = & {\left(\left(\sqrt{1+\delta }\right)d\left(z_{l}^{k^{\prime }},b_{l}^{k^{\prime }}\right)\right)}^{2}=\left(1+\delta \right)d^{2}\left(z_{l}^{k^{\prime }},b_{l}^{k^{\prime }}\right). \end{array}$$Using the inequality (Equation 18), we obtain
(19)$$\begin{array}{ccc} \frac{d\left(z_{l}^{k^{\prime }},b_{l}^{k^{\prime }}\right)+d\left(a_{l}^{k^{\prime }},b_{l}^{k^{\prime }}\right){)}^{2}+1}{d{\left(z_{l}^{k^{\prime }},b_{l}^{k^{\prime }}\right)}^{2}+1}\; & \leq & \frac{\left(1+\delta \right)d{\left(z_{l}^{k^{\prime }},b_{l}^{k^{\prime }}\right)}^{2}+1}{d{\left(z_{l}^{k^{\prime }},b_{l}^{k^{\prime }}\right)}^{2}+1}\\ & \leq & \frac{\left(1+\delta \right)d{\left(z_{l}^{k^{\prime }},b_{l}^{k^{\prime }}\right)}^{2}+1+\delta }{d{\left(z_{l}^{k^{\prime }},b_{l}^{k^{\prime }}\right)}^{2}+1}=1+\delta . \end{array}$$On combining the inequalities (Equation 16) and (Equation 19), we have
(20)$$\frac{d^{2}\left(z_{l}^{k^{\prime }},a_{l}^{k^{\prime }}\right)+1}{d^{2}\left(z_{l}^{k^{\prime }},b_{l}^{k^{\prime }}\right)+1}\leq 1+\delta .$$Again, by the triangle inequality, we have
(21)$$d\left(z_{l}^{k^{\prime }},a_{l}^{k^{\prime }}\right)\geq d\left(z_{l}^{k^{\prime }},b_{l}^{k^{\prime }}\right)-d\left(a_{l}^{k^{\prime }},b_{l}^{k^{\prime }}\right).$$Also, inequality in the assumption (A2a) implies $d\left(z_{l}^{k^{\prime }},b_{l}^{k^{\prime }}\right)-d\left(a_{l}^{k^{\prime }},b_{l}^{k^{\prime }}\right)>0.$Therefore, the inequality (Equation 21) and the positivity of the expression $d\left(z_{l}^{k^{\prime }},b_{l}^{k^{\prime }}\right)-d\left(a_{l}^{k^{\prime }},b_{l}^{k^{\prime }}\right)$ give:
(22)$$\begin{array}{ccc} \frac{d^{2}\left(z_{l}^{k^{\prime }},b_{l}^{k^{\prime }}\right)+\epsilon }{d^{2}\left(z_{l}^{k^{\prime }},a_{l}^{k^{\prime }}\right)+\epsilon }\; & \leq & \frac{d^{2}\left(z_{l}^{k^{\prime }},b_{l}^{k^{\prime }}\right)+\epsilon }{{\left(d\left(z_{l}^{k^{\prime }},b_{l}^{k^{\prime }}\right)-d\left(a_{l}^{k^{\prime }},b_{l}^{k^{\prime }}\right)\right)}^{2}+\epsilon }\leq {\left(\frac{d\left(z_{l}^{k^{\prime }},b_{l}^{k^{\prime }}\right)+\sqrt{\epsilon }}{d\left(z_{l}^{k^{\prime }},b_{l}^{k^{\prime }}\right)-d\left(a_{l}^{k^{\prime }},b_{l}^{k^{\prime }}\right)}\right)}^{2}. \end{array}$$Again, by using the assumption (A2a), we have
$$d\left(a_{l}^{k^{\prime }},b_{l}^{k^{\prime }}\right)\leq \left(1-\frac{1}{\sqrt{2-\delta }}\right)d\left(z_{l}^{k^{\prime }},b_{l}^{k^{\prime }}\right)-\sqrt{\epsilon }.$$On simplifying the above inequality, we obtain
$$\frac{1}{\sqrt{2-\delta }}d\left(z_{l}^{k^{\prime }},b_{l}^{k^{\prime }}\right)+\sqrt{\epsilon }\leq d\left(z_{l}^{k^{\prime }},b_{l}^{k^{\prime }}\right)-d\left(a_{l}^{k^{\prime }},b_{l}^{k^{\prime }}\right).$$Therefore,
$$\begin{array}{ccc} \frac{1}{\sqrt{2-\delta }}d\left(z_{l}^{k^{\prime }},b_{l}^{k^{\prime }}\right)+\frac{\sqrt{\epsilon }}{\sqrt{2-\delta }} & \leq & \frac{1}{\sqrt{2-\delta }}d\left(z_{l}^{k^{\prime }},b_{l}^{k^{\prime }}\right)+\sqrt{\epsilon }\leq d\left(z_{l}^{k^{\prime }},b_{l}^{k^{\prime }}\right)-d\left(a_{l}^{k^{\prime }},b_{l}^{k^{\prime }}\right). \end{array}$$The above inequality gives:
(23)$$\frac{d\left(z_{l}^{k^{\prime }},b_{l}^{k^{\prime }}\right)+\sqrt{\epsilon }}{d\left(z_{l}^{k^{\prime }},b_{l}^{k^{\prime }}\right)-d\left(a_{l}^{k^{\prime }},b_{l}^{k^{\prime }}\right)}\leq \sqrt{2-\delta }.$$Combining inequalities (Equation 22) and (Equation 23), we obtain
(24)$$\frac{d(z_{l}^{k^{\prime }},b_{l}^{k^{\prime }})+\epsilon }{d(z_{l}^{k^{\prime }},a_{l}^{k^{\prime }})+\epsilon }\leq {\left(\sqrt{2-\delta }\right)}^{2}=2-\delta .$$The inequalities (Equation 21) and (Equation 24) give us
(25)$$\Psi_{l}^{k^{\prime }}=\frac{d^{2}\left(z_{l}^{k^{\prime }},a_{l}^{k^{\prime }}\right)+1}{d^{2}\left(z_{l}^{k},b_{l}^{k}\right)+1}\times \frac{d^{2}\left(z_{l}^{k},b_{l}^{k}\right)+\epsilon }{d^{2}\left(z_{l}^{k^{\prime }},a_{l}^{k^{\prime }}\right)+\epsilon }\leq \left(1+\delta \right)\left(2-\delta \right).$$Therefore, by Equations (Equation 9), (Equation 10), and inequalities (Equation 14) and (Equation 25), we have
$$\Psi_{l}^{k}\geq \frac{1}{\left(1+\delta )(2-\delta \right)}\;\mathrm{and}\;\Psi_{l}^{k^{\prime }}\leq \left(1+\delta \right)\left(2-\delta \right).$$Since log is an increasing function, we have
$$\log\Psi_{l}^{k}\geq \log\frac{1}{\left(1+\delta )(2-\delta \right)}\;\mathrm{and}\;\log\Psi_{l}^{k^{\prime }}\leq \log\left(1+\delta \right)\left(2-\delta \right).$$Therefore,
$$\log\Psi_{l}^{k}\geq -\log\left(1+\delta \right)\left(2-\delta \right)\;\mathrm{and}\;-\log\Psi_{l}^{k^{\prime }}\geq -\log\left(1+\delta \right)\left(2-\delta \right).$$By the Equation (Equation 10), we have
(26)$$\begin{array}{ccc} \Delta_{k} & \geq & -\sum_{l=1}^{m^{\prime }}\log\left(1+\delta \right)\left(2-\delta \right)-\frac{1}{r}\sum_{k^{\prime }\neq k}^{}\sum_{l=1}^{m^{\prime }}\log\left(1+\delta \right)\left(2-\delta \right)\\ & \geq & -m^{\prime }\log\left(1+\delta \right)\left(2-\delta \right)-\frac{1}{r}\sum_{k^{\prime }\neq k}^{}m^{\prime }\log\left(1+\delta \right)\left(2-\delta \right)\\ & \geq & -m^{\prime }\log\left(1+\delta \right)\left(2-\delta \right)\left(1+\frac{1}{r}\sum_{k^{\prime }\neq k}^{}\right). \end{array}$$Note that, $\sum_{k^{\prime }\neq k}=n-1$, thus
(27)$$-\Delta_{k}\leq m^{\prime }\log\left(1+\delta \right)\left(2-\delta \right)\left(1+\frac{1}{r}\left(n-1\right)\right).$$The -ve sign indicates the decrease in the logit after the projection of a prototype. □

## 4. Discussion

Ps-ProtoPNet is closely related to three interpretable deep learning models ProtoPNet, Gen-PrtoPNet and NP-ProtoPNet, but strikingly different from them as explained the Section 2.3. Ps-ProtoPNet uses a generalized version of the distance function L2 along with the non-optimization of the last layer. The non-optimization of the last layer helps to preserve the negative connection of the logits with incorrect classes that further helped to establish Theorem 1. Moreover, the non-optimization of the last layer helped us to include the negative reasoning process along with positive reasoning process.

## 5. Conclusions

The non-optimization of the last layer and the use of prototypes with rectangular spatial dimensions and square spatial dimensions greater than 1×1 helped our model to improve its performance over NP-ProtoPNet, Gen-ProtoPNet and ProtoPNet.

## Figures and Tables

**Figure 1 diagnostics-11-01732-f001:**
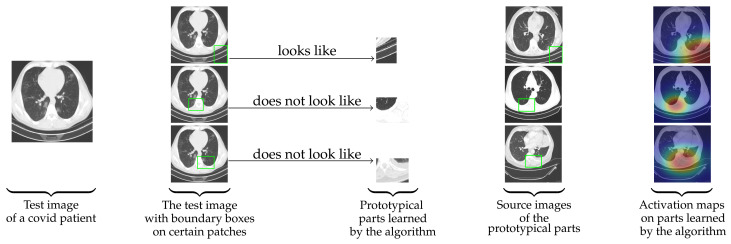
For a given CT-scan image, Ps-ProtoPNet identifies the parts of the image where it thinks that this part of the image looks like that prototypical part, and this part of the image does not look like that prototypical part.

**Figure 2 diagnostics-11-01732-f002:**
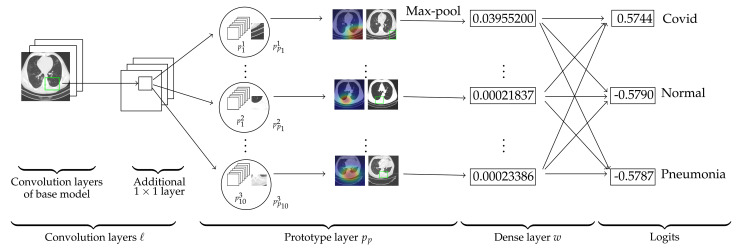
Ps-ProtoPNet architecture.

**Figure 3 diagnostics-11-01732-f003:**
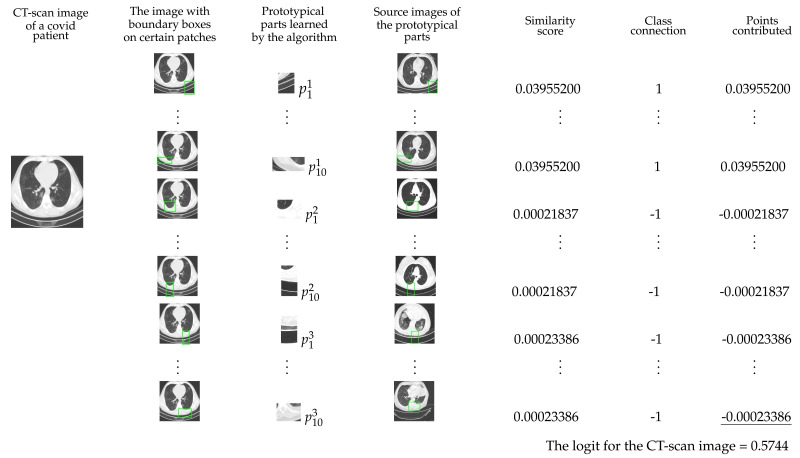
Explanation of the classification process of the model.

**Figure 4 diagnostics-11-01732-f004:**
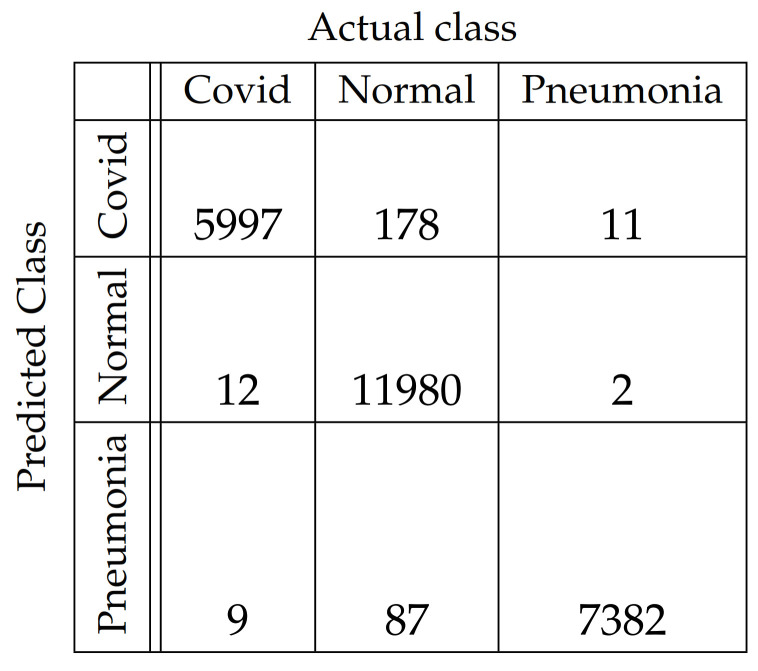
Ps-ProtoPNet with base VGG-16.

**Figure 5 diagnostics-11-01732-f005:**
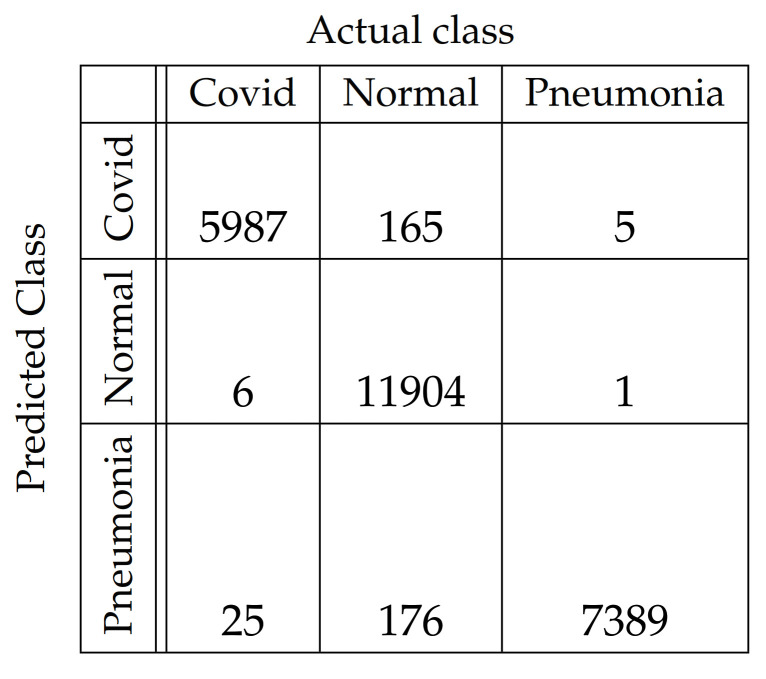
Ps-ProtoPNet with base VGG-19.

**Figure 6 diagnostics-11-01732-f006:**
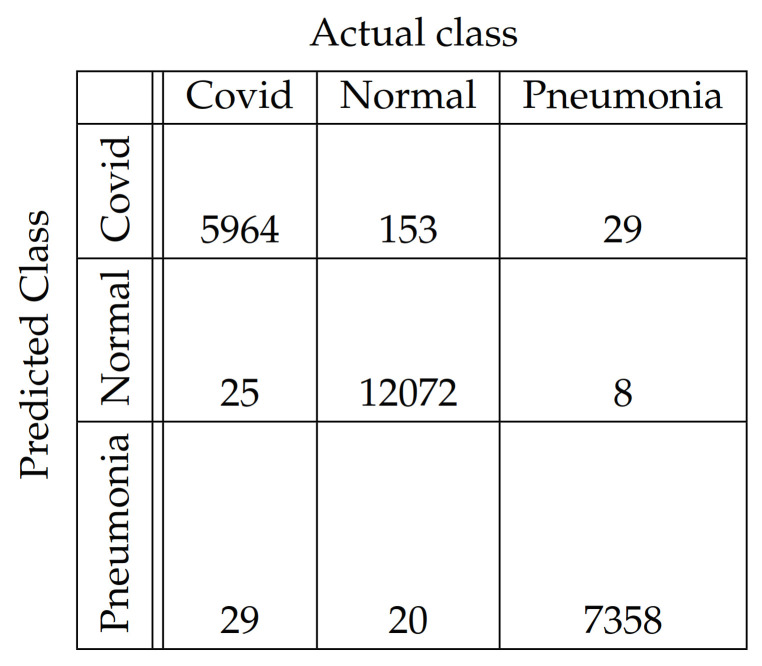
Ps-ProtoPNet with ResNet-34.

**Figure 7 diagnostics-11-01732-f007:**
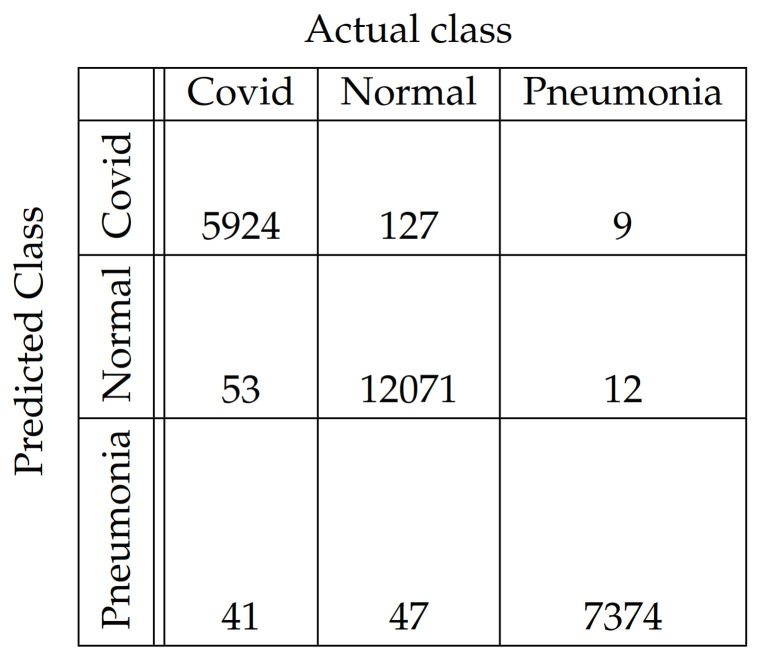
Ps-ProtoPNet with ResNet-152.

**Figure 8 diagnostics-11-01732-f008:**
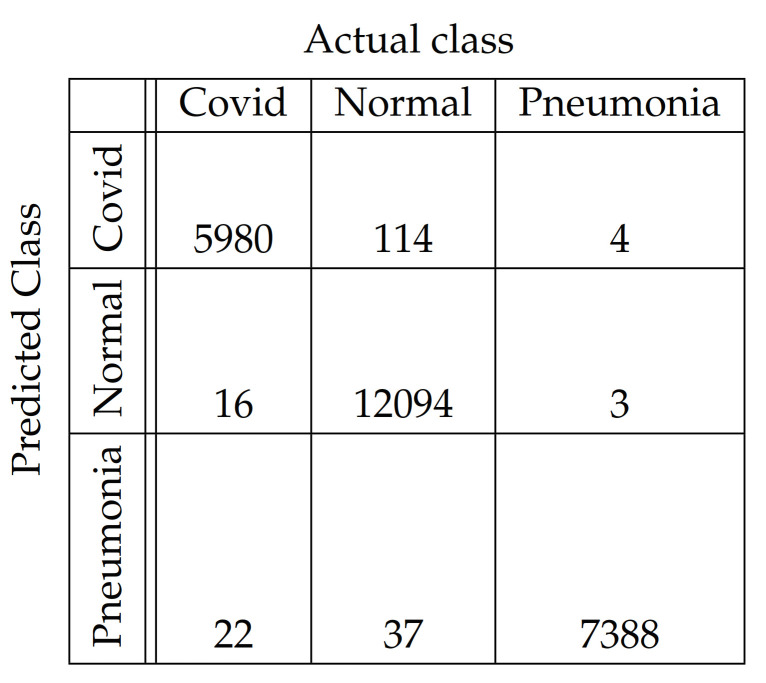
Ps-ProtoPNet with DenseNet-121.

**Figure 9 diagnostics-11-01732-f009:**
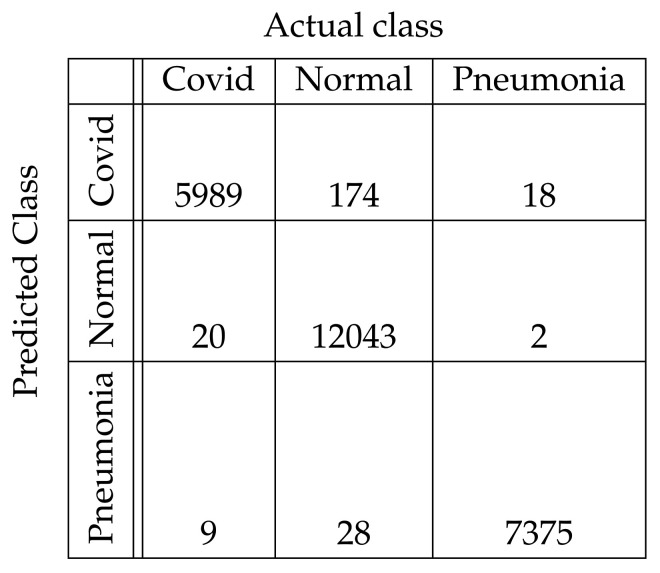
Ps-ProtoPNet with DenseNet-161.

**Figure 10 diagnostics-11-01732-f010:**
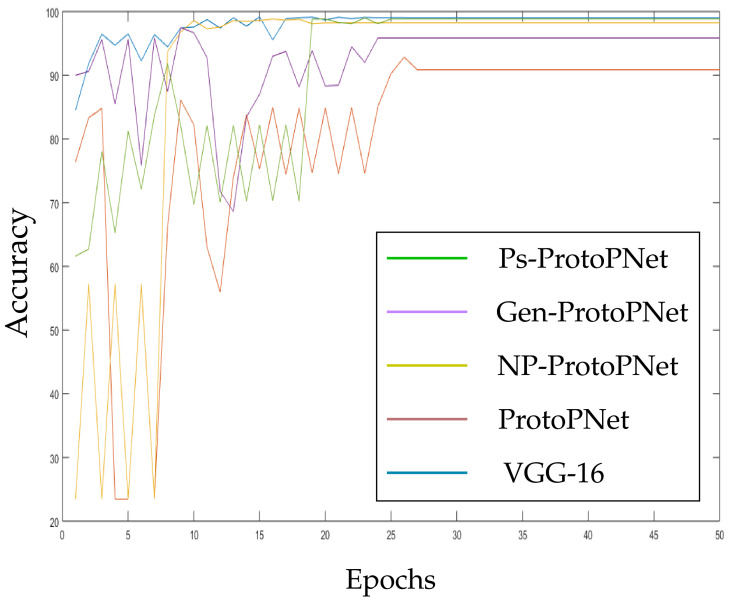
Ps-ProtoPNet with baseline VGG-16.

**Figure 11 diagnostics-11-01732-f011:**
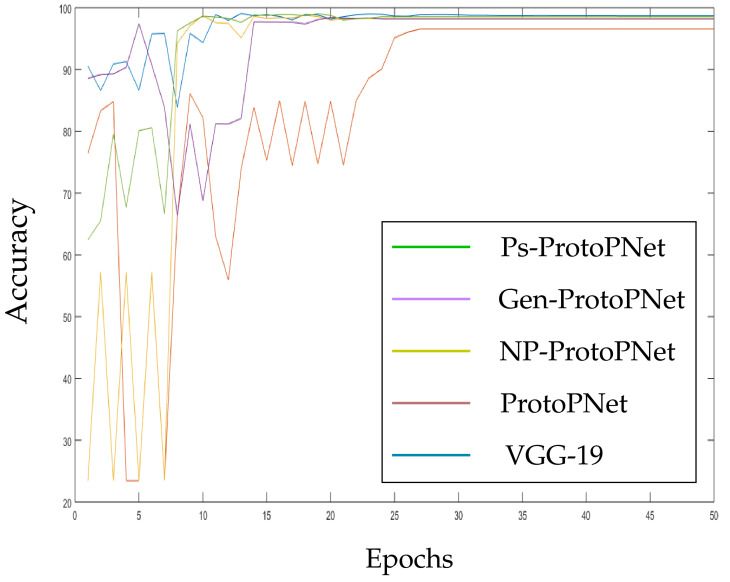
Ps-ProtoPNet with baseline VGG-19.

**Figure 12 diagnostics-11-01732-f012:**
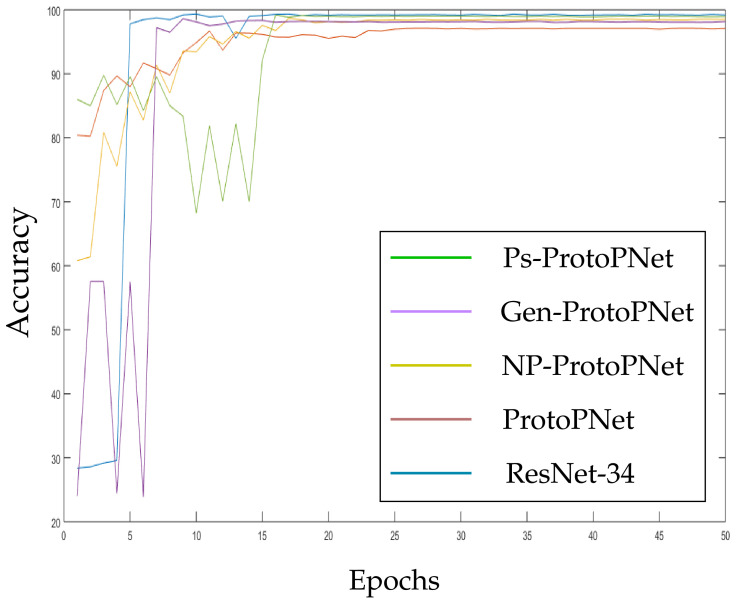
Ps-ProtoPNet with baseline ResNet-34.

**Figure 13 diagnostics-11-01732-f013:**
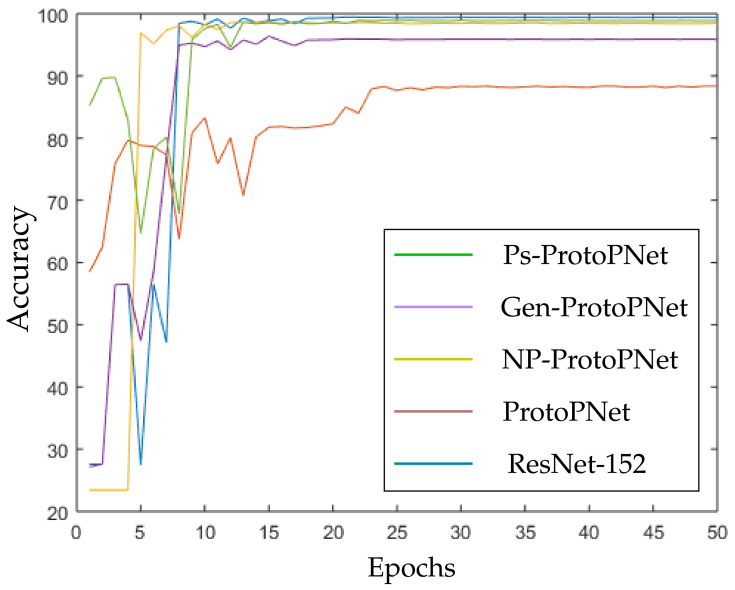
Ps-ProtoPNet with baseline ResNet-152.

**Figure 14 diagnostics-11-01732-f014:**
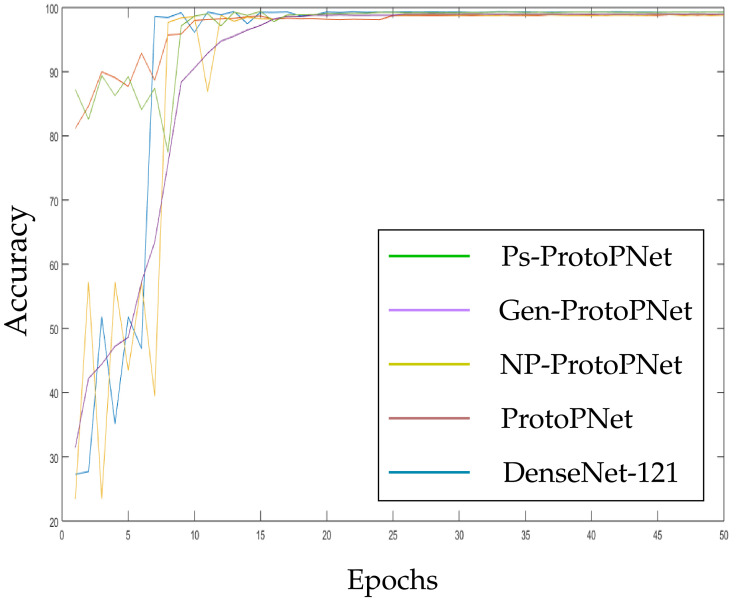
Ps-ProtoPNet with baseline DenseNet-121.

**Figure 15 diagnostics-11-01732-f015:**
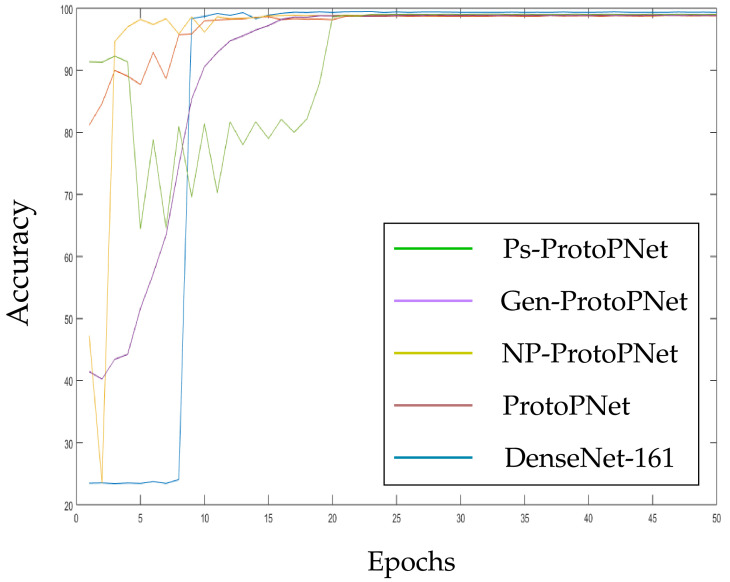
Ps-ProtoPNet with baseline DenseNet-161.

**Table 1 diagnostics-11-01732-t001:** The performances comparison of the models while experimented over the dataset of the CT-scan images.

Base (B)	Metric	Ps-ProtoPNet	Gen-ProtoPNet [14]	NP-ProtoPNet [17]	ProtoPNet [5]	B Only
VGG-16		3 × 4				
	accuracy	98.83	95.85	98.23	90.84	99.03
	precision	0.96	0.93	0.93	0.89	0.98
	recall	0.98	0.95	0.95	0.91	0.99
	F1-score	0.97	0.94	0.94	0.90	0.98
VGG-19		3 × 6				
	accuracy	98.53	98.17	98.23	96.54	98.71
	precision	0.97	0.95	0.91	0.93	0.98
	recall	0.99	0.99	0.96	0.95	0.99
	F1-score	0.98	0.97	0.93	0.94	0.98
ResNet-34		3 × 3				
	accuracy	98.97 ± 0.05	98.40 ± 0.12	98.45 ± 0.07	97.05 ± 0.06	99.24 ± 0.10
	precision	0.97	0.96	0.96	0.95	0.99
	recall	0.99	0.99	0.99	0.96	0.99
	F1-score	0.98	0.97	0.97	0.96	0.99
ResNet-152		2 × 3				
	accuracy	98.85 ± 0.04	95.90 ± 0.09	98.48 ± 0.06	88.20 ± 0.08	99.40 ± 0.05
	precision	0.97	0.93	0.99	0.87	0.99
	recall	0.98	0.93	0.99	0.87	0.99
	F1-score	0.97	0.93	0.99	0.87	0.99
DenseNet-121		3 × 5				
	accuracy	99.24 ± 0.05	98.97± 0.02	98.83 ± 0.10	98.81 ± 0.07	99.32 ± 0.03
	precision	0.98	0.98	0.99	0.98	0.99
	recall	0.99	0.99	0.98	0.98	0.99
	F1-score	0.98	0.98	0.98	0.98	0.99
DenseNet-161		2 × 2				
	accuracy	99.02 ± 0.03	98.87 ± 0.02	98.88 ± 0.03	98.76 ± 0.07	99.41 ± 0.07
	precision	0.96	0.98	0.97	0.97	0.99
	recall	0.99	0.99	0.99	0.99	0.99
	F1-score	0.97	0.98	0.97	0.98	0.99

**Table 2 diagnostics-11-01732-t002:** The *p*-values obtained with the test of hypothesis for system of two proportions (accuracies) between our proposed model, Ps-ProtoPNet, and each of the other model.

Base (B)	Gen-ProtoPNet [17]	NP-ProtoPNet [18]	ProtoPNet [16]	B Only
VGG-16	0.0002	0.0002	0.0002	**0.0367**
VGG-19	0.0007	0.0036	0.0002	**0.0409**
ResNet-34	0.0002	0.0002	0.0002	0.0002
ResNet-152	0.0002	0.0002	0.0002	0.0002
DenseNet-121	0.0002	0.0002	0.0002	**0.0582**
DenseNet-161	**0.0467**	**0.0582**	0.0075	0.0002

## Data Availability

The data used in this study are openly available [44].

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
