# Peer review of "Object or Background: An Interpretable Deep Learning Model for COVID-19 Detection from CT-Scan Images"

_diagnostics, 2021, doi:10.3390/diagnostics11091732_

Round 1
Reviewer 1 Report
The use (or inference) of prototypes in deep learning models is not new. For instance, identifying the prototypes attempting to give some explanation in the diagnostic process through automatic diagnosis is an interesting path; however, claiming explainability is too ambitious: the prototype selection, and the classification layers as well are still somewhat "black-boxed".
Reviewer 2 Report
The manuscript is about an algorithm deep learning-based able to classify CT images as COVID-19 positive, pneumonia, or healthy.
The topic is of interest, even if the manuscript can be improved.
More in details:
- I would mention some COVID-19 CT datasets, such as those in https://www.eibir.org/covid-19-imaging-datasets/ and in “Zaffino P, Marzullo A, Moccia S, Calimeri F, De Momi E, Bertucci B, Arcuri PP, Spadea MF. An Open-Source COVID-19 CT Dataset with Automatic Lung Tissue Classification for Radiomics. Bioengineering. 2021 Feb;8(2):26.” (just to point out some exemplary cases).
- I would move several subsections of the results in materials and methods.
If the subsection describes the method (or a part of it), it would be put in materials and methods. - The authors mention a 1x1 kernel...does it means it is a 2D workflow? What about a 3D strategy?
- It is not clear to me why a tensor 512x7x7 is obtained.
- Please run some statistical tests to figure out if there is statistical difference among models.
In summary: the topic is interesting and the idea looks interesting. Some edits are required.
Reviewer 3 Report
The paper is well written and data presentation is beautiful. I have a few questions as below.
1, I think chest CT scans may be helpful to diagnose COVID-19 with high clinical suspicion of infection. RT-PCR is still gold standard for COVID -19 testing. I would suggest author to have discussion on COVID-19 detection method, like nucleic acid testing and CT scans.
2, How can deep learning approach identify COVID-19 CT images from pneumonia CT image? I would suggest authors have a few sentences to address this in introduction.
Round 2
Reviewer 1 Report
The authors made some changes to their work, improving the previous version of the manuscript. However, there are still some major unsolved issues. In the rebuttal letter, the authors write "[..]our model reasons the way humans describe their own thinking in classification tasks[..]" which is a very strong statement brought through the whole paper, yet not sufficiently supported.
Author Response
We are thankful to the reviewers for reviewing our paper and providing us their valuable suggestions. Our answers to their suggestions/points are as follows.
Point 1: The reviewer asks to explain how our model do the reasoning similar to humans. The comment of the reviewer is as follows:
“However, there are still some major unsolved issues. In the rebuttal letter, the authors write "[..]our model reasons the way humans describe their own thinking in classification tasks[..]" which is a very strong statement brought through the whole paper, yet not sufficiently supported.”
Response 1: We agree that our claim "[..]our model reasons the way humans describe their own thinking in classification tasks[..]" in the rebuttal letter is too strong, it should have worded more appropriately as follows:
"[..] the reasoning process of our model is inspired from the way humans describe their own thinking in classification tasks[..]"
To classify images, we might focus on parts of the image and compare them with prototypical parts of images from various classes. This type of reasoning is usually used in difficult identification tasks, such as, identification of unfamiliar images.
For example, radiologists may compare suspected tumor in X-ray or CT-scan image with prototypical tumor images for diagnosis of cancer.
Another example, we also use this type of reasoning to identify the people. Typically, a person has some physical features similar to his parents and even to his siblings. We can say with some certainty that a person belongs to a given family if we remember the faces of the person’s family members (so that we can compare them with the person), because we can compare eyes, nose, lips and some other physical features of the person with the corresponding features of his family members. Similarly, to classify an image we might focus on parts of the image and compare them with prototypical parts of images from given classes. This type of human reasoning is inspired our model. Comparison of image parts with learned prototypes is integral to the reasoning process of the model.
We include the following sentences near the end of the last paragraph of the section “Introduction”:
“To identify an image that has not been encountered before, humans may compare patches of the image with the patches of images of the known objects. Our model’s reasoning is inspired from the above reasoning, where comparison of image parts with learned prototypes is integral to the reasoning process of the model.”

Reviewer 2 Report
The authors addressed all my questions, except one.
In my opinion, a statistical test is needed to verify if the outputs from the different models are statistically different or not.
